# Unraveling the Pomegranate Genome: Comprehensive Analysis of R2R3-MYB Transcription Factors

**Heming Suo [1], Xuan Zhang [1], Lei Hu [1], Huihui Ni [1], Renzeng Langjia [2], Fangyu Yuan [1], Maowen Zhang [1] and Shuiming Zhang [1,*]**

[1]  Department of Ornamental Horticulture, School of Horticulture, Anhui Agricultural University, Hefei 230036, China; shm1220073202@163.com (H.S.); xuan_xuan1211@163.com (X.Z.); l191563178@163.com (L.H.); nhh1477852456@163.com (H.N.); yuanfangyug@163.com (F.Y.)
[2]  Forest Science Research Institute of Tibet, Lhasa 850000, China; rzenglangjia@163.com
*   Correspondence: zhangshuiming@ahau.edu.cn

**Abstract:** R2R3-MYB TFs represent one of the most extensive gene families in plants and play a crucial role in regulating plant development, metabolite accumulation, and defense responses. Nevertheless, there has been no systematic investigation into the pomegranate R2R3-MYB family. In this study, 186 R2R3-MYB genes were identified from the pomegranate genome and grouped into 34 subgroups based on phylogenetic analysis. Gene structure analysis showed that the PgR2R3-MYB family in the same subgroup had a similar structure. Gene duplication event analysis revealed that the amplification of the PgMYB family was driven by Whole Genome Duplication (WGD) and dispersed duplication. In the upstream promoter sequence of the PgMYB gene, we identified a large number of plant hormones and environmental response elements. Using phylogenetic analysis and RNA-seq analysis, we identified three PgMYB TFs that may be involved in the regulation of lignin synthesis. Their expression patterns were verified by qPCR experiments. This study provides a solid foundation for further studies on the function of the R2R3-MYB gene and the molecular mechanism of lignin synthesis.

**Keywords:** lignin; genome-wide identification; phylogenetic analysis; gene expression

## 1. Introduction

Pomegranate (*Punica granatum*) is an ancient cultivar that originated in Central Asia and spread throughout the world. There are two species of *Punica L.* The *P. granatum* is widely distributed throughout the world, while *Punica protopunica Balf. f.* is only found in certain regions. Pomegranates have been cultivated and evolved in China for more than 2000 years to form numerous varieties with rich genetic diversity [1,2]. Pomegranate is generally considered a healthy fruit [3]. Pomegranate seeds are rich in nutrients, including organic acids, sugars, minerals, and many other nutrients [4,5]. They are widely used in traditional medicine in the United States, Asia, Africa, and Europe to treat different types of diseases and have high edible and medicinal value [6]. Consumers prefer soft-seed pomegranate in market, which encourages investigators and breeders to progress and breed soft-seed cultivars [7].

The formation of pomegranate seed hardness is the process of continuous lignin accumulation in the endocarp cells and gradual thickening of the secondary cell wall, which is also the process of lignification of the endocarp cell wall [8]. Lignin, as the main component of the cell wall, interconnects with lignocellulose and hemicellulose to form a meshwork structure that forms the skeleton of plant cells [9]. Niu et al., pointed out that the seed coat cell wall thickness of hard-seeded pomegranate was significantly greater than that of soft-seeded pomegranate [10]. It has also been shown that pomegranate seed hardness is significantly and positively correlated with its lignin content [11,12].

Transcription factors (TFs) regulate gene expression by binding to distal and local cis elements of target genes. TFs are regulatory factors that play important roles in plant development, cell cycle, cell signaling and response to adversity. Typical TFs include transcriptional regulatory regions, nuclear localization signaling regions, oligomerization response regions, and DNA binding regions, and these functional regions determine the structure and characteristics of TFs [13]. Common TFs in plants are WRKY, ERF, bZIP, MYB, and other families, out of which the MYB family is one of the largest TFs families in plants. The MYB gene family is widely present in eukaryotic cells and is one of the largest in plants. The MYB protein consists of 1–4 highly conserved imperfect repeats (R1, R2, and R3). The MYB gene can be divided into four different types, namely 2R (R2R3-MYB), 3R (R1R2R3-MYB), 4R (R1R2R2R1/2-MYB), and 1R-MYB (MYB-related proteins), of which the R2R3-MYB gene accounts for the larger proportion [14]. R2R3-MYB is probably the TF that most directly regulates lignin biosynthesis and deposition, and it can regulate the expression of genes involved in the synthesis of phenylpropane substances, thereby affecting the content of lignin [15,16]. R2R3-MYB TFs play an important role in plant growth and development, such as regulating biological and abiotic stresses and affecting the synthesis of lignin and anthocyanins [17].

In recent years, research of the R2R3-MYB gene has made great progress, and members of this family have been identified in Arabidopsis [18], grape [19], kiwifruit [20], watermelon [21], Pitaya [22], Rhodiola [16], and apples [23]. According to previous studies, in cotton, GhMYB4 could impede lignin deposition in the cell wall by directly binding to and negatively regulating the expression of genes such as GhC4H-1/2, Gh4CL-4, GhCAD-3, and GhLac1, and the reduction of lignin content in GhMYB4 overexpressing cotton leads to enhanced cell wall permeability [24]. The Arabidopsis R2R3-MYB gene family was divided into 24 subgroups [18]. AtMYB58, AtMYB63, and AtMYB85 were TFs specific for lignin biosynthesis in Arabidopsis [25]. AtMYB46 and AtMYB83 were not only regulators of the lignin synthesis pathway but also effectively activated the entire process of secondary cell wall formation [26,27]. The overexpression of the OsMYB91 gene increased the ABA content in the plant and enhanced the ability of rice to resist drought stress [28]. MaMYB13 induced cold resistance by regulating the synthesis of VLCFAs and phenylpropane compounds in bananas [29].

The birth of new cultivars of fruit crops takes a long time, so the identification of related genes is an important work [3,30]. The purpose of this study was to search for lignin-related MYB family members and understand their expression patterns, so as to provide theoretical basis for breeding better soft-seed pomegranate cultivars. For this purpose, the pomegranate R2R3-MYB gene family was identified based on the pomegranate genome. We constructed phylogenetic trees of these genes, conducted intraspecific and interspecific collinearity analyses, and analyzed the structure and promoter cis-acting elements of the PgMYB family. In addition, an evolutionary tree was constructed using MYB members from other species known to regulate lignin synthesis to identify the MYB most likely to be involved in pomegranate lignin synthesis. Further, we selected soft-seed 'Tunisia' and hard-seed cultivars 'Hongyushizi' and 'Baiyushizi' (the lowest lignin content was found in seeds of Tunisia [31]) and drew an expression heatmap with their transcriptomic data, and the genes concerned were verified by qRT-PCR.

## 2. Materials and Methods

### 2.1. Plant Material

In this study, samples were collected from three selected cultivars: hard-seed pomegranates 'Hongyushizi' and 'Baiyushizi' and soft-seed pomegranate 'Tunisia'. They were nine-year-old cultivars planted in the experimental base of Anhui Agricultural University. Samples were collected 40, 80, and 120 days after anthesis, and three biological replicates were set for each sample. The seeds and arils were immediately extracted, placed in liquid nitrogen, and finally frozen in the laboratory at −80 °C.

*2.2. Identification of R2R3-MYB Gene Family Members and Physicochemical Properties Analysis in Pomegranate*

Some 126 Arabidopsis R2R2-MYB protein sequences were downloaded from TAIR (https://www.arabidopsis.org/, assessed on 22 January 2023). They were used as query sequences for blastp analysis of pomegranate genome and extracted sequences with E-value $< 1 \times 10^{-10}$ (other parameters default). To verify our results, the online tool Pfam was used to screen candidate PgMYB sequences to identify MYB domains. CD-search (https://www.ncbi.nlm.nih.gov/Structure/cdd/wrpsb.cgi, assessed on 23 January 2023) was used to view MYB protein domains. Sequences that did not contain the full domain of R2R3-MYB or incomplete reading regions were deleted. Ultimately, 186 members of the PgMYB gene family were identified. The ProtParam tool of Expasy (https://web.expasy.org/protparam, assessed on 25 January 2023) was used to analyze the physical and chemical properties. Online tool Euk-mPLoc 2.0 (http://www.csbio.sjtu.edu.cn/bioinf/euk-multi-2/, assessed on 25 January 2023) was used to predict protein subcellular localization.

*2.3. Phylogenetic Analysis*

To understand the evolutionary relationship of PgMYB, this study concentrated members of the MYB gene family of Arabidopsis and pomegranate and aligned these sequences using the ClustalW function (default parameters) of MGEA7. The study used the sequence alignment results to construct evolutionary trees in MEGA7 [32], utilizing the Neighbor Joining method. The bootstrap replication was set to 1000, and the parameters were default. Finally, in order to better display the results, the study imported the generated evolutionary tree into the online software ITOL (https://itol.embl.de/, assessed on 2 February 2023) for visualization [33]. In order to further screen PgMYB involved in lignin synthesis, we selected validated genes from chrysanthemum, flax, eucalyptus, populus, and Arabidopsis (CmMYB8 [34], EgMYB1 [35], LuMYB12, LuMYB113, LuMYB146 [36], PotMYB216 [37], and AtMYB4 [18]) to construct phylogenetic trees with alternative PgMYB genes.

*2.4. Chromosomal Position and Collinearily Analysis*

The genomes and annotation files of pomegranate, eucalyptus, and Arabidopsis were obtained from NCBI, and collinear analysis was performed using the MCScanX module of TBtools [38]. The TBtools service was used to perform chromosome localization and visualization.

*2.5. Gene Structure and Protein Motif Analysis*

The PgMYB protein sequences identified were used to construct phylogenetic trees. The intron, exon, and genomic localization information of the MYB family were all derived from NCBI (https://www.ncbi.nlm.nih.gov/, assessed on 5 February 2023). The online software MEME (http://meme-suite.org/tools/meme, assessed on 3 February 2023) was used to predict the motif of the pomegranate R2R3-MYB protein, and the number of predictions was set to 10 in the parameters, and the other parameters were set as the default parameters. TBtools was used for visualization.

*2.6. Analysis of Promoter Cis-Acting Elements*

The study obtained the 2000 bp upstream sequences of the PgMYB gene by the Gtf/GFF3 Sequences Extractor option in TBtools. The Plant CARE (http://bioinformatics.psb.ugent.be/webtools/plantcare/html/, assessed on 3 February 2023) service was used to perform promoter homeopathic element analysis. Finally, we used Excel to visualize the result.

*2.7. RNA Extraction Library Construction, and Sequencing*

Total RNA was extracted from samples using a Trizol reagent kit (Invitrogen, Carlsbad, CA, USA) according to the manufacturer's protocol. RNA quality was assessed on an

Agilent 2100 Bioanalyzer (Agilent Technologies, Palo Alto, CA, USA) and checked using RNase free agarose gel electrophoresis. After total RNA was extracted, we used Oligo(dT) beads to enrich mRNA, and prokaryotic mRNA was enriched by removing rRNA by a RiboZeroTM Magnetic Kit (Epicentre, Madison, WI, USA). Then, the enriched mRNA was fragmented into short fragments by using a fragmentation buffer and reversely transcribed into cDNA using a NEBNext Ultra RNA Library Prep Kit for Illumina (NEB #7530, New England Biolabs, Ipswich, MA, USA). The purified double-stranded cDNA fragments were end repaired, with a base added, and ligated to Illumina sequencing adapters. The ligation reaction was purified with the AMPure XP beads (1.0X). Ligated fragments were subjected to size selection by agarose gel electrophoresis and polymerase chain reaction (PCR) amplification. The resulting cDNA library was sequenced using Illumina Novaseq6000 by Gene Denovo Biotechnology Co. (Guangzhou, China).

### 2.8. Expression of R2R3-MYB Genes in Pomegranate

To obtain high quality clean reads, reads were filtered by fastp [39] (version 0.18.0). Short reads alignment tool Bowtie2 [40] (version 2.2.8) was used for mapping reads to the ribosome RNA (rRNA) database. The rRNA mapped reads then were removed. The remaining clean reads were further used in assembly and gene abundance calculation. Paired-end clean reads were mapped to the reference genome using HISAT2.2.4 [41]. The mapped reads of each sample were assembled by using StringTie v1.3.1 [42,43] in a reference-based approach. For each transcription region, a FPKM (fragment per kilobase of transcript per million mapped reads) value was calculated to quantify its expression abundance and variations, using RSEM software [44]. Finally, TBtools was used for visualization.

### 2.9. Real-Time Quantitative PCR (qRT-PCR)

Total RNA was extracted using the TRIzol® Plus RNA Purification Kit (Invitrogen, Carlsbad, CA, USA). The cDNA was synthesized using the SuperScript™ III First-Strand Synthesis SuperMix for qRT-PCR (Invitrogen, Carlsbad, United States) according to the manufacturer's protocol. Specific primers were designed by Beacon Designer 7.8 software (Table S1). qRT-PCR was performed with Power SYBR® Green PCR Master Mix (Roche, Basel, Switzerland). The reaction system is shown in Table S2. Three technical replicates were performed for each sample.

## 3. Results

### 3.1. Identification of R2R3-MYB Gene Family Members Physicochemical Properties of Pomegranate R2R3-MYB Proteins

In this study, 186 PgMYBs were identified from the pomegranate genome. To identify all members of the R2R3-MYB gene family in pomegranates, blastp was used to search, and a total of 206 candidate genes were obtained. This study removed 20 sequences that do not contain the full domain of R2R3-MYB or incomplete reading regions, and a total of 186 R2R3-MYB genes, called MYB1-MYB186, were obtained. The physicochemical properties of the 186 sequences were analyzed (Table S3), and the results showed that the amino acid length of the Pomegranate MYB gene family was between 94 (MYB100) and 1516 (MYB174). This is a large range, but most proteins do not differ greatly in length. The molecular weight of the proteins was 10.55 kDa–169.43 kDa, and the isoelectric point (pI) range was 4.81 (MYB96)–9.96 (MYB20), of which 100 proteins were acidic and pI was less than 7. Instability coefficients ranged from 39.19 (MYB103) to 73.97 (MYB2). The Aliphatic index was between 50.89 (MYB115) and 90.16 (MYB22). All PgMYBs were hydrophobic, and their average hydrophilicity was less than 0. Subcellular localization of 186 MYB sequences in pomegranate was predicted, with 184 structures localized on the nucleus and 19 on the cytoplasm. There were two localizations on the cytomembrane and one localization on the extracellular matrix.

### 3.2. Phylogenetic Analysis of the R2R3-MYB Gene Family

The Arabidopsis MYB family had been studied deeply, and many MYB functions had been verified. Therefore, researchers can infer the function of the same subgroup of genes from the phylogenetic tree. This study constructed phylogenetic trees with the R2R3-MYB gene families of *Arabidopsis thaliana* and pomegranate (Figure 1). In Dubos's study, the Arabidopsis R2R3-MYB gene family was divided into 25 subgroups [18]. The study referred to his classification result and divided the R2R3-MYB gene family of pomegranate into 34 subgroups, ensuring that each MYBs were classified. There are 31 subgroups of PgMYBs and AtMYBs, but a few members fail to cluster with AtMYBs. These branches that do not contain AtMYB are S26, S27, S28, and S32. The constructed phylogenetic tree not only facilitates the exploration of phylogenetic relationships within the PgMYB family but also enables identification of lignin-related PgMYBs, given that numerous AtMYB genes have been functionally characterized.

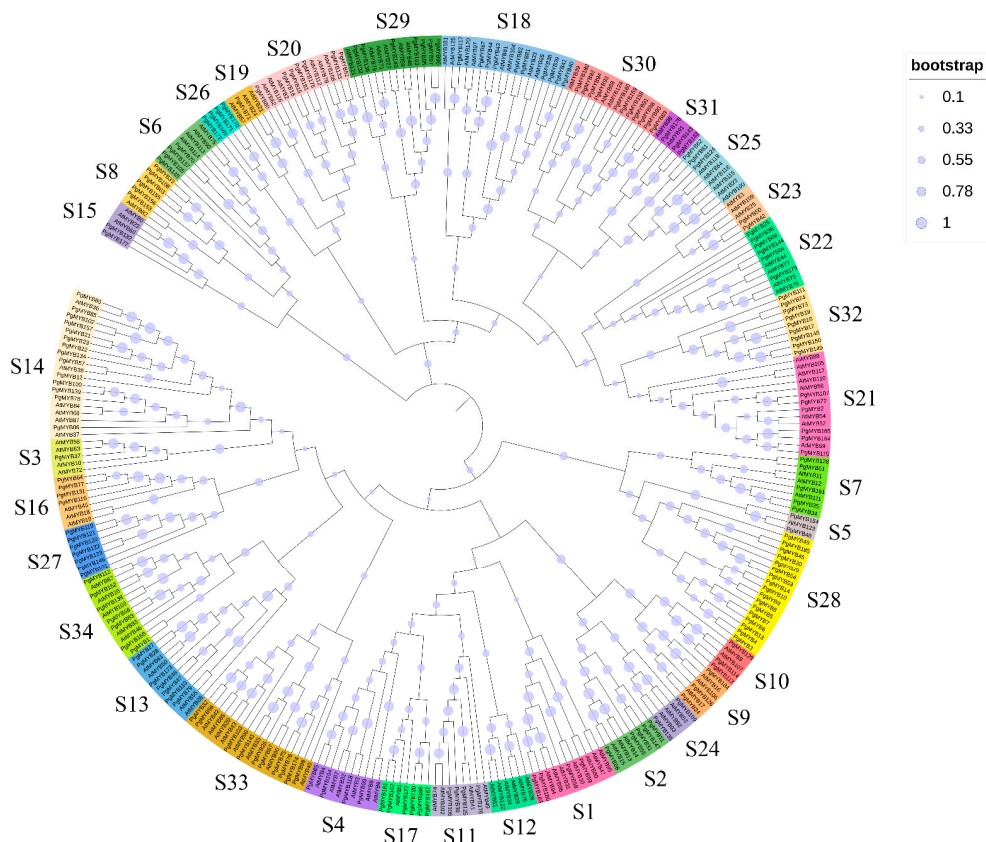

**Figure 1.** Phylogenetic relationships of R2R3-MYB gene family in pomegranate. The color blocks on the right indicate the different subgroups. The blue circle represents the bootstrap value: the larger the circle, the higher the bootstrap value.

Arabidopsis S3, S13, S21, and AtMYB46 and AtMYB83 have all been verified to be involved in lignin biosynthesis. Since the homologous genes usually have similar functions, the function of the corresponding orthologous genes in pomegranate can be inferred. Therefore, PgMYBs in the same subgroup as S3, S13, and S21 of Arabidopsis thaliana (PgMYB2, PgMYB37, PgMYB27, PgMYB28, PgMYB46, PgMYB47, PgMYB72, PgMYB79, PgMYB110, PgMYB107, PgMYB115, PgMYB164, PgMYB165, and PgMYB173) and homologous genes PgMYB66 and PgMYB83 of AtMYB46 and AtMYB83 may be involved in lignin biosynthesis. In order to further study the regulatory relationship of PgMYBs on lignin synthesis, the study selected validated genes from chrysanthemum, flax, eucalyptus, Populus, and Arabidopsis to construct phylogenetic trees with alternative PgMYB genes. The results showed that (Figure 2) nine of the PgMYBs (PgMYB37, PgMYB2,

PgMYB115, PgMYB165, PgMYB66, PgMYB83, PgMYB173, PgMYB110, and PgMYB79) were closely related, suggesting that these genes might have similar functions.

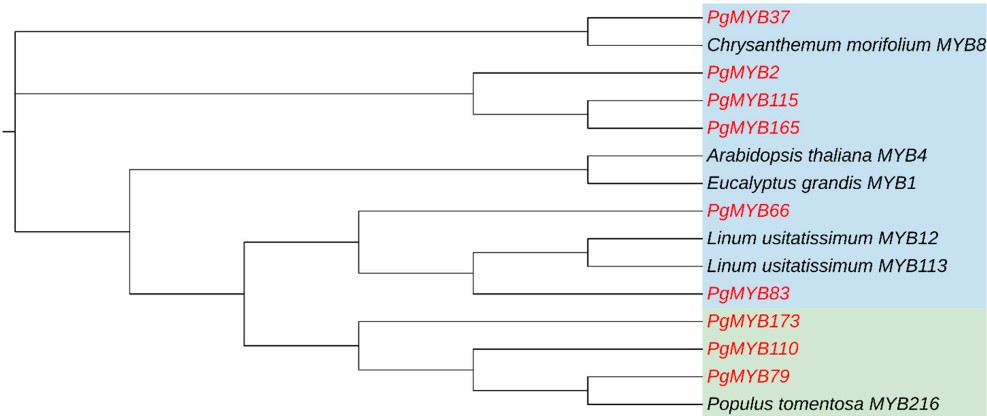

**Figure 2.** Phylogenetic analysis of Lignin-related PgMYB. The pomegranate MYB genes are highlighted in red.

### 3.3. Chromosomal Position and Collinearity Analysis

The pomegranate has a total of eight chromosomes, and 186 PGMYBs are unevenly distributed among them (Figure 3). In the figure, PgMYBs were concentrated in areas of the chromosome with high gene density. Chromosome 1 contains the most PgMYB genes (33 genes, 17.7%), followed by chromosome 4 (29 genes, 15.6%). Chromosome 6 contains the least amount of PgMYB genes (16 genes, 8.6%). Through collinearity analysis, 45 collinear gene pairs were found in PgMYB family.

To further investigate the potential evolutionary mechanisms of the R2R3-MYB subfamily, a comparative syntenic map was constructed based on pomegranate, *Arabidopsis*, and *Eucalyptus grandis* (Figure 4). This study found 138 orthologous gene pairs between pomegranate and *Arabidopsis thaliana* and 173 orthologous gene pairs between pomegranate and *Eucalyptus grandis*. This means that there is a closer evolutionary distance between eucalyptus and pomegranate than pomegranate and Arabidopsis. Further, MCScanX was used to detect duplication events in the pomegranate MYB gene family (Table S4). There are five duplication events in genes: WGD, dispersed, tandem, proximal, and singleton. This study found four of these duplication events in 145 PgMYBs (the duplication events of 41 genes remained undiscovered) except for singleton. Specifically, 53.10% (77) of PgMYBs were retained from WGD events. Secondly, 37.24% (56) of PgMYBs were from dispersed events. Only 6.20% (9) were from tandem events, and 2.06% (3) were from proximal events.

### 3.4. R2R3-MYB Gene Structures and Protein Domains Analysis in Pomegranate

To further elucidate the evolutionary relationship between pomegranate MYB genes, the phylogenetic tree of 186 PgMYB proteins was constructed (Figure 5a). In addition, the conservative motifs of the PgMYB proteins were predicted by the online software MEME (Figure 5b). This study found significant differences in the number of motifs for PgMYB TFs. The most complex genes were PgMYB21, PgMYB22, and PgMYB23, all containing nine motifs, while the simplest gene was PgMYB96, containing only Motif2. Motif1 (97.3%), Motif2 (96.8%), Motif3 (98.9%), and Motif4 (78.1%) are commonly found in PgMYB TFs. Motif5 was found in 40.6% of members, and Motif6 was found in 23% of members. Only a small subset of members contains Motif7 (5.3%), Motif8 (4.3%), Motif9 (5.3%), and Motif10 (4.3%). Members of the same subgroup in the evolutionary tree showed similar conserved motifs. For example, the members of the S19 subgroup all included Motif1, Motif2, Motif3, and Motif4, and the members of the S25 subgroup all included Motif1, Motif2, Motif3, and Motif8.

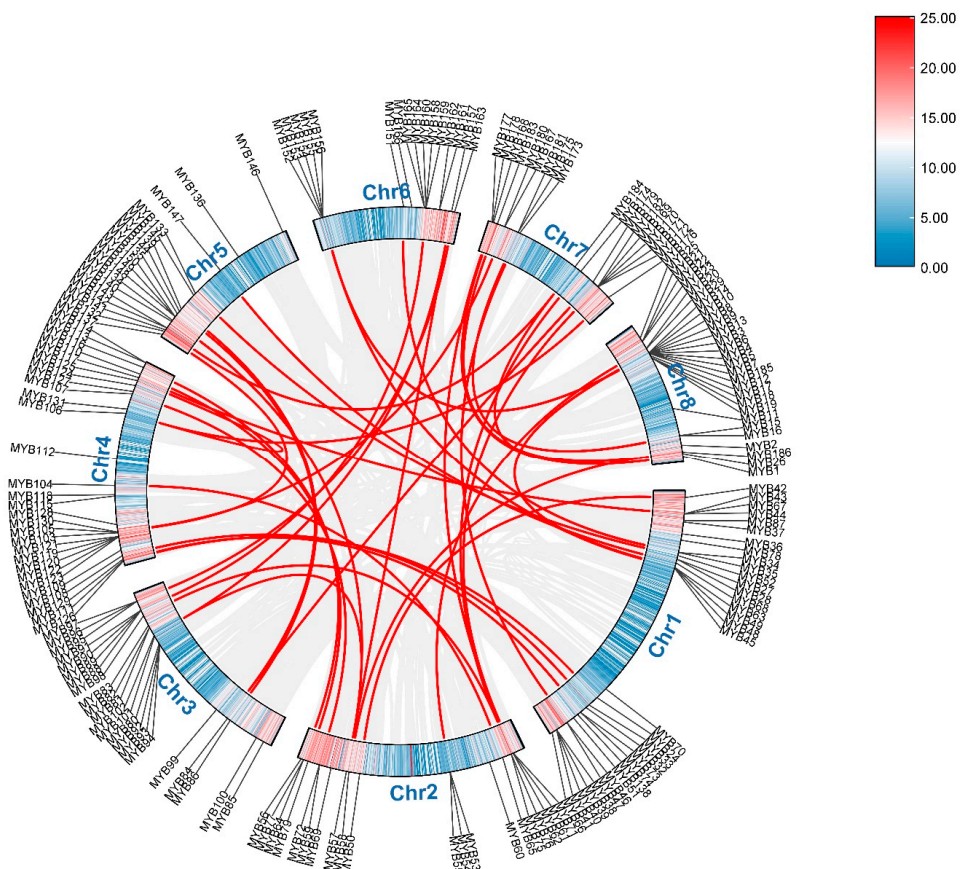

**Figure 3.** Chromosomal distribution of PgMYB genes. The color bands on the chromosomes indicate gene density, with red indicating high gene density and blue indicating low gene density. The red lines connect paralogous genes of PgMYB.

This study visualized the gene structure of the pomegranate R2R3-MYB family (Figure 5c). The phylogenetic development of the PgMYB gene family showed that the sequences of the same subgroup had similar exon and intron structures, but the overall results showed differences in the number of introns and exons in the pomegranate R2R3-MYB gene family. Even though there were significant differences in the length of MYB genes in pomegranate, there was still some regularity in their genetic structure. This study found that 147 members had three exons and 28 members had two or four exons. The gene structure containing two introns was common in the PgMYB family, where it was found in 150 members. PgMYB88, PgMYB89, PgMYB90, and PgMYB91 had 12 exons and 11 introns, the largest number of exons and introns of any member. In addition, PgMYB21 and PgMYB138 had eight exons and seven introns, PgMYB17, PgMYB18, and PgMYB19 had 11 exons and 10 introns.

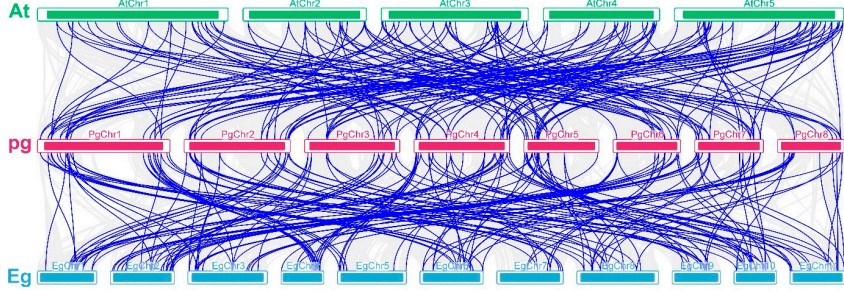

**Figure 4.** Genome-wide synteny analysis for R2R3-MYB genes among *Punica granatum*, *Eucalyptus grandis*, and *Arabidopsis thaliana*. The blue lines indicate ortholog gene pairs. Eg, *Eucalyptus grandis*; Pg, *Punica granatum*; At, *Arabidopsis thaliana*.

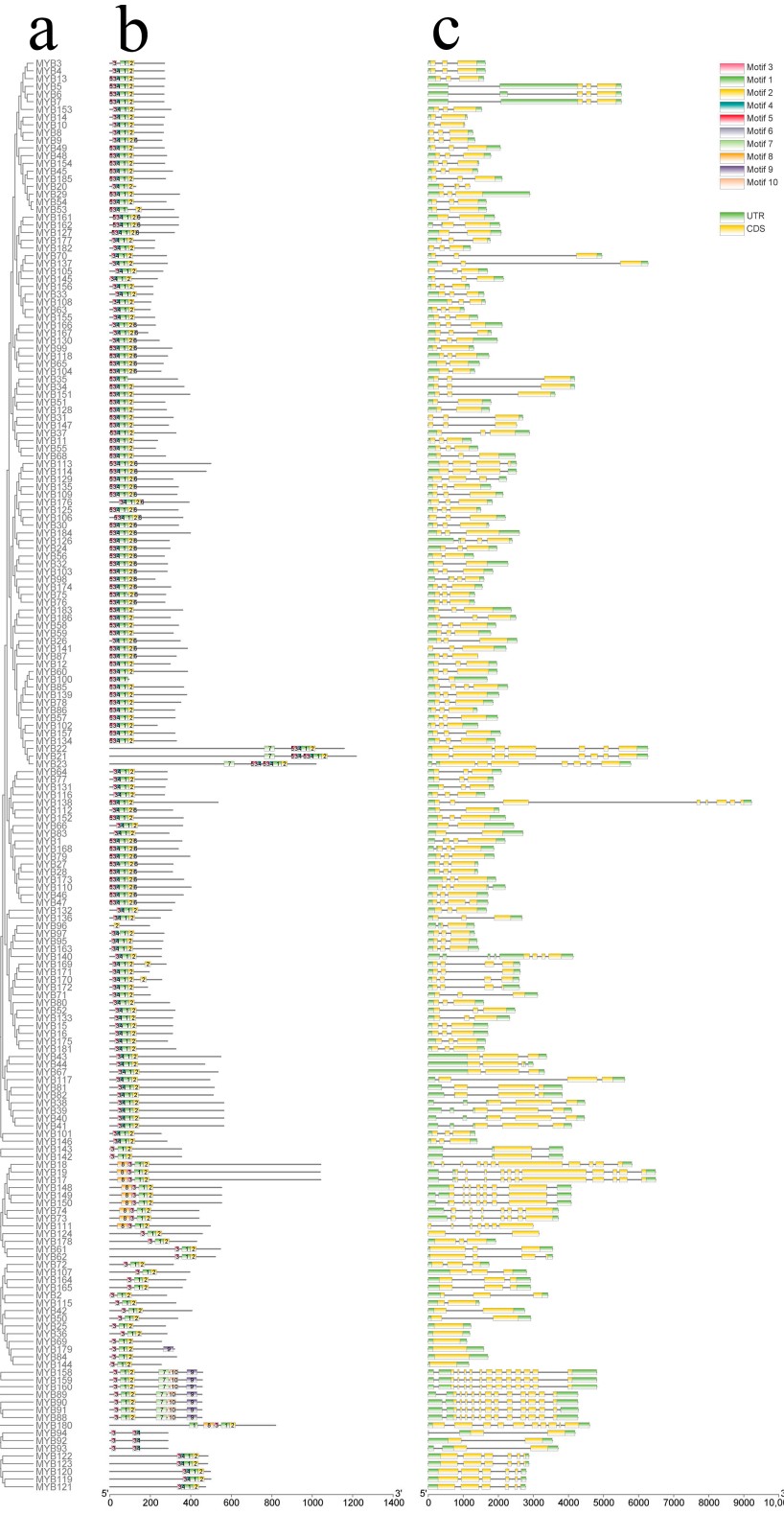

**Figure 5.** The gene structures and protein domains of PgMYB members, (**a**): phylogenetic tree of PgMYBs; (**b**): Protein motif of PgMYBs, they were named according to the E-value of the motif. The scale at the bottom indicates the sequence length. (**c**): Gene structures of PgMYBs, Green boxes indicated UTR, Yellow boxes indicated CDS, and gray lines indicated introns. UTR: untranslated region, CDS: Coding sequence.

### 3.5. Promoter Analysis of R2R3-MYB Genes in Pomegranate

To investigate the potential function of the pomegranate R2R3-MYB gene family, the study analyzed the 2000 bp sequence upstream of the PgMYB gene family CDS. A large number of cis-acting elements were detected (Figure 6). These elements were classified as promoter elements, plant growth elements, and environmental response elements. This study revealed the ubiquitous presence of ABRE, TGACE-motif, and CGTCA-motif in PgMYB promoter sequences, which played a crucial role in abscisic acid and methyl jasmonate responses. It indicates that MYB family generally had a hormone regulation effect. In addition, PgMYB exhibited a high prevalence of ARE, GC-motifs, and MBS elements which were implicated in low temperature, anaerobic, and drought stress responses. Suggesting the general responsiveness of PgMYB to abiotic stresses, it is likely that this transcription factor plays a crucial role in plant resistance against such environmental challenges.

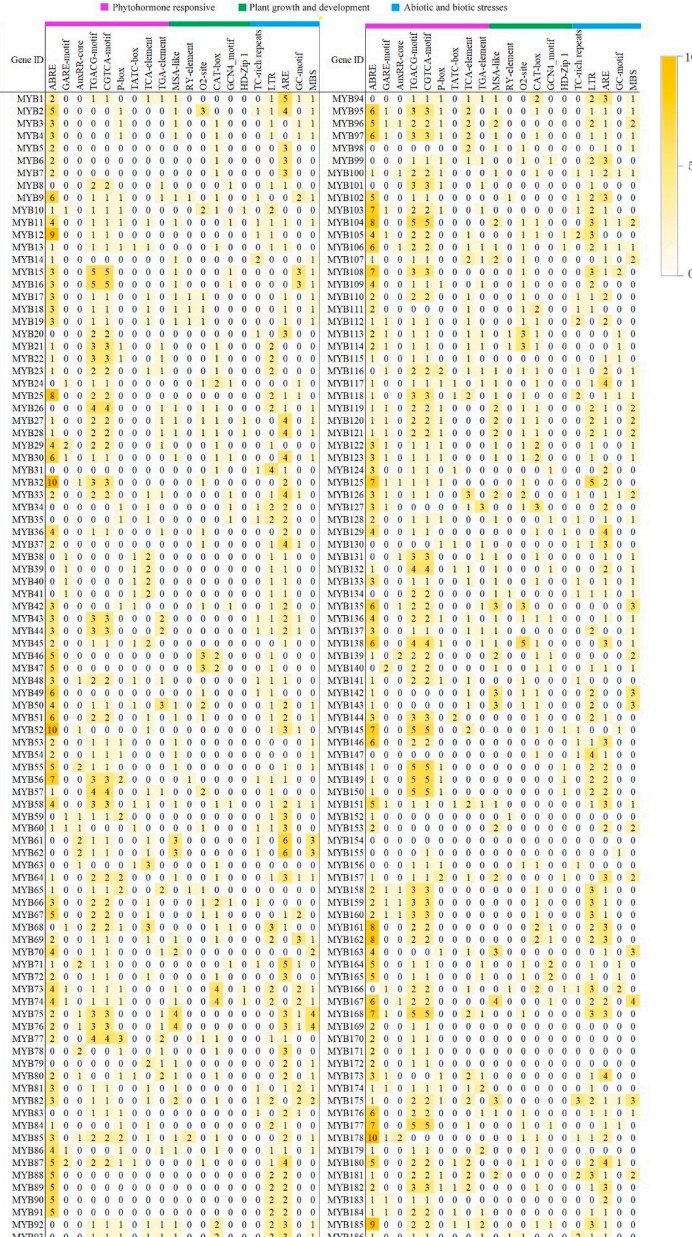

**Figure 6.** Number of cis-acting elements in the promoter region of PgMYB genes. The more cis-acting elements are predicted, the darker the colors appear in the diagram.

### 3.6. Expression of R2R3-MYB Genes

In phylogenetic analysis, a screening process was conducted to identify nine PgMYB genes that potentially participate in lignin biosynthesis. The expression information of these genes was obtained from RNA-seq data in this study (Table S5). The temporal (Figure 7a–c) and interspecific (Figure 7d–f) expression patterns of these genes were visualized using six heatmaps. Notably, PgMYB66 was conspicuously absent from the transcriptome data. Given that the transcriptome data utilized in this study were obtained from pomegranate seeds, PgMYB66 may be expressed in other organs.

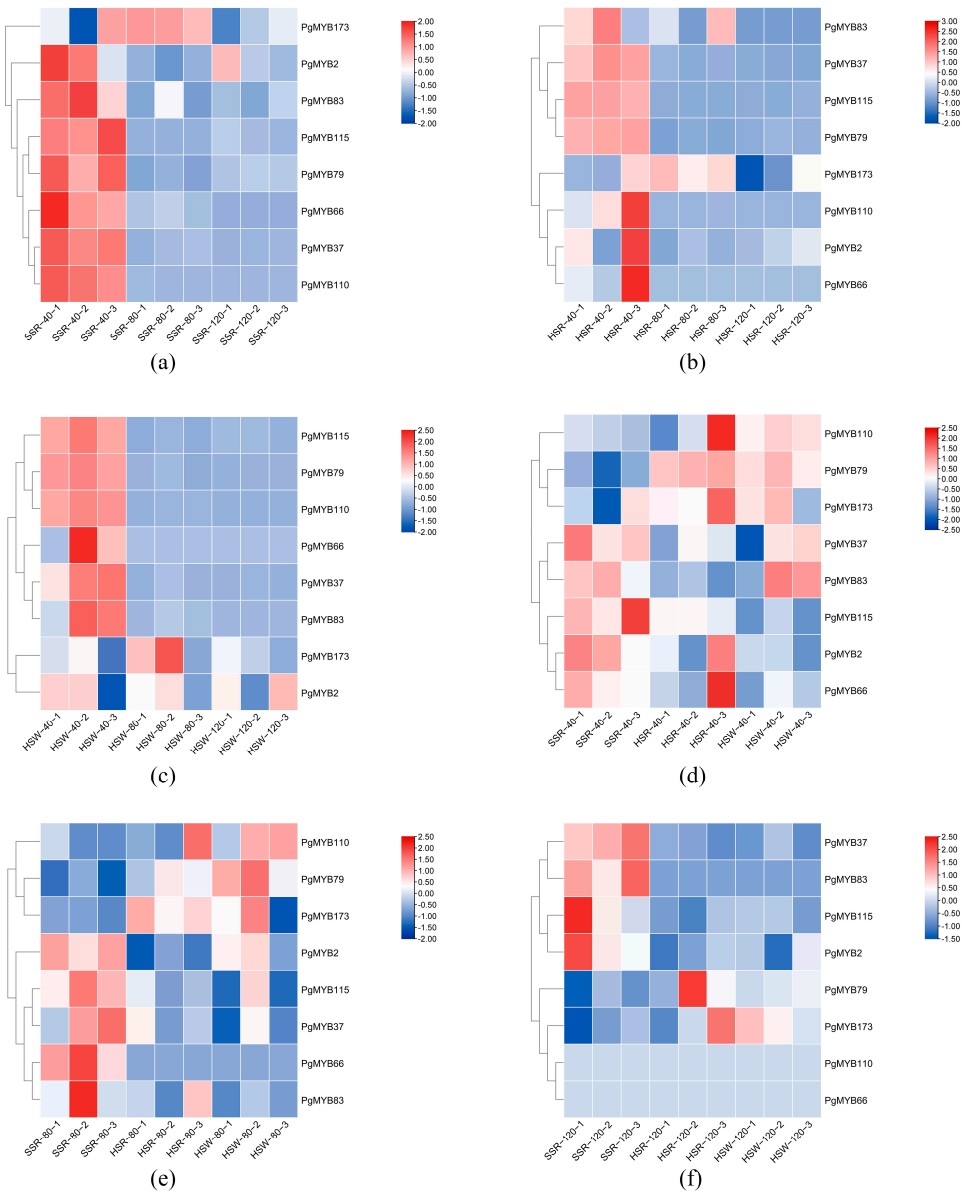

**Figure 7.** Expression patterns of PgMYB in pomegranate seeds. (**a**) Expression patterns of PgMYB at different developmental stages in Tunisia. (**b**) Expression patterns of PgMYB at different developmental stages in Hongyushizi. (**c**) Expression patterns of PgMYB at different developmental stages in Baiyushizi. (**d**) PgMYB expression patterns of different cultivars at 40 days after flowering. (**e**) Expression patterns of PgMYB in different cultivars at 80 days after flowering. (**f**) Expression patterns of PgMYB in different cultivars at 120 days after flowering. The scale on the left of the picture represents expression levels, with red indicating high expression and blue indicating low expression. SSR: Soft-seed cultivar "Tunisia"; HSW: hard-seed cultivar "Baiyushizi"; HSR: hard-seed cultivar "Hongyushizi".

Temporal expression analysis of the selected PgMYB genes (Figure 7a–c) indicated that their highest expression levels were observed in seeds 40 days after flowering. Notably, PgMYB173 exhibited the greatest level of expression among the three cultivars at 80 days after anthesis, while no significant differences were detected in the expression patterns of PgMYB2 during various stages of Baiyushizi (Figure 7c) seed development. By comparing the expression patterns of different cultivars simultaneously (Figure 7d–f), this study found similar expression patterns of these genes in different cultivars 40 (Figure 7d) days and 80 days (Figure 7e) after flowering. PgMYB110, PgMYB79, and PgMYB173 were highly expressed in Hongyushizi and Baiyushizi. PgMYB37, PgMYB83, PgMYB115, PgMYB2, and PgMYB66 were more highly expressed in Tunisia. PgMYB37, PgMYB83, PgMYB115, PgMYB2, and PgMYB66 were more highly expressed in Tunisia at 120 days after flowering (Figure 7f). PgMYB79 and PgMYB173 were highly expressed in Hongyushizi and Baiyushizi. PgMYB110 and PgMYB66 were expressed at low to no levels at 120 days after flowering. Notably, the expression of PgMYB2 and PgMYB115 in hard-seed cultivars (Hongyushizi and Baiyushizi) had been maintained at low levels, and PgMYB79 had been maintained at high levels throughout development.

### 3.7. The Validation of PgMYB Expression with qRT-PCR

The expressions of PgMYB2, PgMYB79, and PgMYB115 in Tunisia and Hongyushizi were detected by qRT-PCR (Figure 8). The results showed that the expression levels of all three were highest at 40 days after anthesis, decreased gradually with fruit development, and reached the lowest at 120 days after anthesis. Compared with the hard-seed cultivar 'Hongyushizi', PgMYB2 and PgMYB115 had higher expression in the soft-seed cultivar 'Tunisia', and PgMYB79 had higher expression in 'Hongyushizi'. The expression of all three genes exhibited a consistent temporal pattern, with the highest level observed at 40 days after flowering and a subsequent gradual decline. The expression pattern is similar to the heatmap.

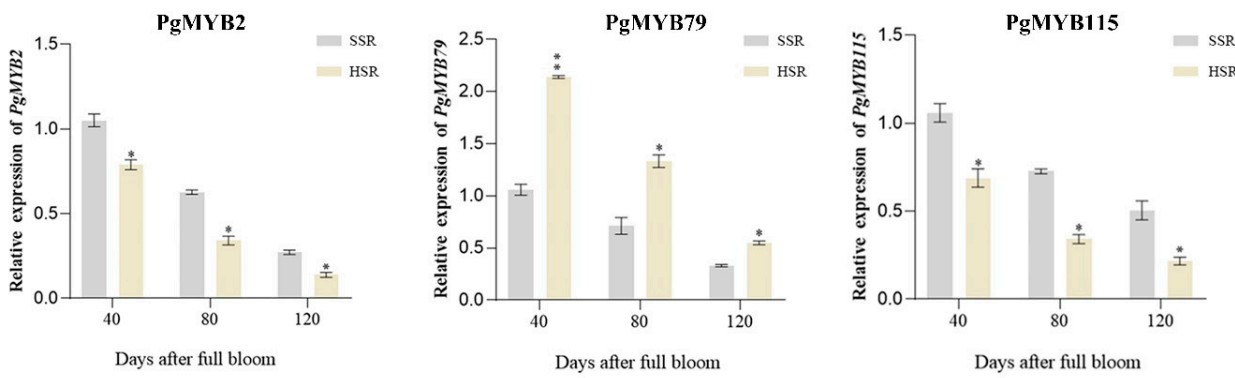

**Figure 8.** The validation of PgMYB Expression with qRT-PCR. The gray column represents 'Tunisia' (SSR) expression level tested by qRT-PCR; the yellow column represents 'Hongyushizi' (HSR) expression level; error bars indicate the standard deviation of qRT-PCR data. Asterisks indicate those values that are significantly different from the control (* $p < 0.05$, ** $p < 0.01$).

## 4. Discussion

Pomegranate has garnered significant attention from both consumers and researchers alike [45,46], owing to its exceptional nutritional, medicinal, antioxidant, and antibacterial properties [47,48]. However, the poor taste caused by the hardness of pomegranate seeds hinders further market expansion [3]. Lignin plays a pivotal role in the development of pomegranate seed hardness traits [12]. Current research on lignin biosynthesis in pomegranate has primarily focused on key enzymes that directly regulate lignin synthesis, with limited attention given to transcription factors. In this study, the pomegranate MYB transcription factor family was comprehensively analyzed for the first time. A total of 186 PgMYB genes were identified from the pomegranate genome, and their characteristics

and structures were predicted and analyzed. Furthermore, MYB TFs potentially involved in lignin biosynthesis were identified through the utilization of phylogenetic tree and RNA-seq data.

Lignin is widely present in the plant body and is an important component of vascular plants' secondary cell walls, providing mechanical strength to resist damage from external factors [49]. MYB TFs are involved in lignin and cellulose biosynthesis and deposition and have important regulatory roles in secondary cell wall formation [50]. EgMYB2, the homologous gene of AtMYB46 in (*Eucalyptus grandis W. Mill ex Maiden*), binds to AC elements located on lignin synthetase promoters to regulate lignin synthesis [51]. Both maize ZmMYB167 [52] and poplar (*Populus tomentosa*) PtoMYB216 [53] have been shown to involve MYB TFs in lignin biosynthesis. This shows that the higher the homology between MYB class transcription factor members, the greater the similarity of their functions. Therefore, the functional characteristics of other MYB class transcription factor family members on the same branch can be predicted by the existing functional study of MYB class transcription factor family members in Arabidopsis.

In our study, PgMYB family was divided into 34 subgroups, which is consistent with the findings of Hou et al. [54]. The phylogenetic trees of pomegranate and Arabidopsis thaliana showed that most MYBs were in the same branch as Arabidopsis, indicating that pomegranate and Arabidopsis R2R3-MYB members had similar evolutionary origins, but S12 lacked pomegranate MYB members and S26, S27, S28, and S32 lacked Arabidopsis MYB members. Abundant evidence from previous studies has shown that the R2R3-MYB family is extensively involved in phenylpropane metabolism and regulates lignin synthesis [55,56]. Based on functional genes in *Arabidopsis thaliana*, S3, S13, and S21 were thought to be involved in the regulation of lignin synthesis. Based on homology, nine of these genes were found to be more likely involved in the regulation of lignin biosynthesis.

R2R3-MYB is a vast gene family with significant variations in membership properties and structure [57], which also holds true for pomegranate. In this study, the smallest PgMYB transcription factor comprises only 94 amino acids, while the largest one consists of 1516 amino acids. Pomegranate is a diploid plant species (2n = 16) [58], consisting of eight chromosomes per haploid set. The MYB gene has been identified on each chromosome. Notably, the quantity of R2R3-MYB genes in plant genomes does not necessarily correlate with genome size or ploidy level [57]. Collinear analysis showed that the MYB family had expanded extensively during evolution. There were 138 orthologous gene pairs between pomegranate and Arabidopsis and 173 orthologous gene pairs between pomegranate and Eucalyptus grandis. Due to the extensive replication of MYB TFs during evolution, new members are involved in specific functions [16]. In this study, large-scale WGD or segmental and dispersed duplication were detected in PgMYBs. Whole genome duplication is an important evolutionary origin in early plants [59], while dispersed duplication may have originated from transposable elements [60]. WGD and dispersed duplication are the main driving forces of MYB family amplification in pomegranate. The same phenomenon was found in the study of Li et al. [61].

In this study, PgMYB TFs belonging to the same subgroup exhibited identical motif structures. Similarly, genes in the same subgroup typically exhibit the same pattern of introns and exons, including the location and number of introns. These results suggest that the pattern of introns in pomegranates is highly conserved rather than random. In addition, the majority of PgMYB genes exhibit a typical splicing pattern consisting of three exons and two introns, which is consistent with observations in other plant species. We analyzed the cis-acting elements in the upstream sequence of PgMYB genes and found that they contain a large number of plant hormone response elements, which can be induced by light, drought, cold, and other environmental factors. In addition, we found MYB-binding sites upstream of certain PgMYB genes, indicating that these members might cooperate with other members to perform or enhance function.

Gene expression patterns provide important clues for gene function. We utilized RNA-seq data from nine candidate genes to produce expression heatmaps showing their

expression patterns during fruit development in different cultivars. The results showed that the expression of PgMYB79 in hard-seed cultivars was higher than that in soft-seed cultivars at all stages, indicating that PgMYB79 was involved in the positive regulation of lignin synthesis. The expressions of PgMYB2 and PgMYB115 were higher in soft-seed cultivars, suggesting that PgMYB2 and PgMYB115 were involved in the negative regulation of lignin synthesis. Therefore, we employed qPCR to validate their expression profiles, which exhibited a general consistent pattern with the heatmaps.

**5. Conclusions**

In this study, 186 R2R3-MYB TF members were identified from the pomegranate genome, and a series of bioinformatics analyses were performed to reveal their characteristics. Furthermore, we screened PgMYB members that may participate in regulating lignin biosynthesis. This study provides valuable information for further studies on MYB and lignin synthesis.

**Supplementary Materials:** The following supporting information can be downloaded at: https://www.mdpi.com/article/10.3390/horticulturae9070779/s1, Table S1: Primer sequences and target genes for qPCR analysis; Table S2. qRT-PCR reaction system; Table S3: Basic information of the PgMYB gene family; Table S4: Duplication events of PgMYB; Table S5: Expression data of PgMYB.

**Author Contributions:** Conceptualization, X.Z.; Data curation, H.N. and L.H.; software, H.S. and M.Z.; formal analysis, X.Z. and H.S.; Writing—original draft, X.Z. and H.S.; Writing—review and editing, S.Z.; visualization, R.L., H.N. and F.Y.; supervision, S.Z.; funding acquisition, S.Z. All authors have read and agreed to the published version of the manuscript.

**Funding:** This research and the APC were funded by the Natural Science Foundation of Anhui Province, grant number 2008085MC100, and the Anhui Provincial Natural Science Research Project Fund, grant number KJ2019ZD19.

**Data Availability Statement:** The original transcriptome data used in this study have been submitted to the NCBI, and the data are stored in the SRA database. the accession number is "PRJNA914887".

**Conflicts of Interest:** The authors declare no conflict of interest.

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
