# Peer review of "Unraveling the Pomegranate Genome: Comprehensive Analysis of R2R3-MYB Transcription Factors"

_horticulturae, doi:10.3390/horticulturae9070779_

Round 1
Reviewer 1 Report
It is suggested that the manuscript needs language editing.
The title is excellent and appropriate for the proposed scientific research
The article is too long with more of the author's work and reads like a review with many details.
The abstract and conclusion are very generic and should summarise the purpose and what is presented as new within the paper.
The aim is not clear in the introduction as well as the significance of the paper is not highlighted
The neatness of the paper must be improved.
There are many paragraphs, some of which are long and some of which are short, which appear to be disorderly.
Methodology requires to be well clarified
Please make all tables self-explanatory.
The discussion should be further improved.
Check your references
This paper can be accepted with minor revisions, but I must revise the paper once more to ensure that all my comments were incorporated.

It is suggested that the manuscript needs language editing.
Reviewer 2 Report
In the current study, the Authors identified and analyzed the R2R3-MYB gene family in pomegranate cultivars to understand their association with lignin biosynthesis. While employing transcriptomic and gene expression approaches in soft and hard seed cultivars, the authors have suggested the potential involvement of R2R3MYB members in lignin biosynthesis. This manuscript presents intriguing results accompanied by high-quality figures. However, I have identified several issues and areas of ambiguity that need to be addressed before considering acceptance of this manuscript.
Authors compared the transcriptomic data of soft and hard seed cultivars and generated heat maps of MYBs. However, there is a lack of molecular insight into lignin biosynthetic genes in both cultivars. Therefore, I suggest that the authors include a comparative heat map of genes/enzymes related to the lignin biosynthesis pathway and qPCR validation in these cultivars.
Authors should provide the parameters used for generating the alignment and phylogenetic trees. Additionally, the bootstrap values or scale are missing in phylogenetic trees.
The number of biological replicates used in the study to generate transcriptomic/expression data is not clear. Please specify this information.
Could you provide more detail on how the expression data were extracted from the transcriptomic data? Additionally, it would be helpful to include the heatmap data (expression) and corresponding statistical analysis in the supplementary data.
Details regarding the statistical tools used for analyzing the expression data are missing!
It is recommended to move the tables to the supplementary data section.
NA
Reviewer 3 Report
The manuscript is interesting and it is in general well writing. The information presented in MandM section and results is a little bit disorder, the items should be presented in the same order in both sections. MandM section should be greatly improved and more details should be provided. The discussion is the most poor section of the manuscript, results should be discussed better, besides more information about other papers should be mentioned. In addition, all results obtained in the present paper should be discussed.
Several comments and suggestions were added to the document.

Some minor comments about some sentences were added to the manuscript.
Round 2
Reviewer 1 Report
I have read your manuscript, “Unraveling the Pomegranate Genome: Comprehensive Analysis of R2R3-MYB Transcription Factors.” The revisions that you made to the manuscript are very effective in addressing the remaining concerns. my decision is to accept the manuscript for publication in Horticulturae journal.
Thank you for your patience with the review process.
Reviewer 2 Report
Thank you for the revised manuscript, it seems the authors have addressed the concerns raised in the current version.